# α-Ketoglutarate-Mediated DNA Demethylation Sustains T-Acute Lymphoblastic Leukemia upon TCA Cycle Targeting

**DOI:** 10.3390/cancers14122983

**Published:** 2022-06-16

**Authors:** Yanwu Wang, Ning Shen, Gervase Spurlin, Sovannarith Korm, Sarah Huang, Nicole M. Anderson, Leah N. Huiting, Hudan Liu, Hui Feng

**Affiliations:** 1Taikang Medical School (School of Basic Medical Science), Wuhan University, Wuhan 430071, China; wangyanwu@whu.edu.cn; 2Department of Pharmacology and Experimental Therapeutics, Boston University School of Medicine, Boston, MA 02118, USA; shenning6666@gmail.com (N.S.); gspurlin@bu.edu (G.S.); ksnarith07@gmail.com (S.K.); sarahh00@bu.edu (S.H.); nicand@pennmedicine.upenn.edu (N.M.A.); lhuiting@bu.edu (L.N.H.); 3Department of Medicine, Section of Hematology and Medical Oncology, Boston University School of Medicine, Boston, MA 02118, USA; 4Department of Hematology, Zhongnan Hospital of Wuhan University, Wuhan 430071, China; hudanliu@whu.edu.cn; 5Frontier Science Center for Immunology and Metabolism, Medical Research Institute, Wuhan University, Wuhan 430071, China

**Keywords:** TCA cycle, T-cell acute lymphoblastic leukemia, α-ketoglutarate, DNA demethylation, oxidative phosphorylation, reductive carboxylation

## Abstract

**Simple Summary:**

A promising anti-cancer strategy is to target the tumor’s dependence on particular nutrients. However, resistance to single-agent treatment is common due to the adaptability of cancer cells. This research seeks to understand how T acute lymphoblastic leukemia (T-ALL) escapes disruption of the tricarboxylic acid (TCA) cycle, a biochemical pathway critical for T-ALL survival. We show that leukemic cells modify their DNA to increase the activity of other pathways to compensate for diminished TCA cycle function. Our findings will help guide the rational selection of therapies to overcome drug resistance.

**Abstract:**

Despite the development of metabolism-based therapy for a variety of malignancies, resistance to single-agent treatment is common due to the metabolic plasticity of cancer cells. Improved understanding of how malignant cells rewire metabolic pathways can guide the rational selection of combination therapy to circumvent drug resistance. Here, we show that human T-ALL cells shift their metabolism from oxidative decarboxylation to reductive carboxylation when the TCA cycle is disrupted. The α-ketoglutarate dehydrogenase complex (KGDHC) in the TCA cycle regulates oxidative decarboxylation by converting α-ketoglutarate (α-KG) to succinyl-CoA, while isocitrate dehydrogenase (IDH) 1 and 2 govern reductive carboxylation. Metabolomics flux analysis of T-ALL reveals enhanced reductive carboxylation upon genetic depletion of the E2 subunit of KGDHC, dihydrolipoamide-succinyl transferase (DLST), mimicking pharmacological inhibition of the complex. Mechanistically, KGDHC dysfunction causes increased demethylation of nuclear DNA by α-KG-dependent dioxygenases (e.g., TET demethylases), leading to increased production of both IDH1 and 2. Consequently, dual pharmacologic inhibition of the TCA cycle and TET demethylases demonstrates additive efficacy in reducing the tumor burden in zebrafish xenografts. These findings provide mechanistic insights into how T-ALL develops resistance to drugs targeting the TCA cycle and therapeutic strategies to overcome this resistance.

## 1. Introduction

Metabolic reprogramming is a hallmark of cancer that can be therapeutically exploited [1,2,3]. The oncogenic transcription factor MYC drives the rewiring of multiple metabolic pathways in a broad spectrum of cancers, enhancing glycolysis, glutaminolysis, oxidative phosphorylation (OXPHOS), and nucleotide and amino acid synthesis [4,5,6,7]. T-cell acute lymphoblastic leukemia (T-ALL) is an aggressive hematologic malignancy of developing thymocytes that often harbors gain-of-function mutations of *NOTCH1*, leading to aberrant activation of its downstream gene *MYC* [8,9].

Different from normal T cells, transformed T-ALL cells depend on the tricarboxylic acid (TCA) cycle for growth and survival [4,10,11]. The TCA cycle is a central node in carbon metabolism, providing leukemic cells with substrates for macromolecular synthesis to fuel uncontrolled proliferation. However, the demand for cycle substrates is often not met by glucose-derived acetyl-CoA [12,13]. MYC-driven T-ALL thus becomes addicted to glutamine and utilizes it as an alternative carbon input for the TCA cycle [14].

Glutamine anaplerosis enables tumor cells to replenish TCA cycle intermediates via the conversion of glutamine to α-KG [12,15,16]. Glutamine is first converted to glutamate and then to α-KG, which is irreversibly decarboxylated by the α-ketoglutarate dehydrogenase complex (KGDHC) to succinyl-coA [4]. Alternatively, α-KG can be converted into isocitrate by isocitrate dehydrogenase (IDH) and then to citrate via NADPH-dependent reductive carboxylation [17]. Of the three isoforms of IDH, IDH1 and IDH2 located in the cytosol and mitochondrial matrix, respectively, regulate reductive carboxylation in a cancer-specific manner [18].

Approaches to directly target glutamine anaplerosis have been challenging, as the pharmacologic inhibition of glutaminases with glutamine analogs or competitive inhibitors frequently causes excessive tissue toxicity [19,20,21,22,23]. CB-839 is an allosteric glutaminase inhibitor, which presents less toxicity but only leads to moderate anti-tumor efficacy as a single-agent treatment [21,23]. Conversely, directly targeting the TCA cycle is gaining more attention as a potential treatment strategy, as it can inhibit the downstream use of glutamine-derived carbon skeletons, in addition to disrupting cellular bioenergetic and redox homeostasis [4].

This approach is also intriguing in light of our recent demonstration that high expression of the TCA cycle transferase DLST contributes to tumor development in multiple MYC-driven cancers, such as T-ALL, high-risk neuroblastoma, and triple-negative breast cancer (TNBC) [11,20,24,25,26,27,28]. CPI-613, which targets mitochondrial KGDHC and pyruvate dehydrogenase, is an example of the TCA cycle inhibitors currently in clinical trials for treatment of a broad spectrum of cancers, including acute myelogenous leukemia (AML) and pancreatic adenocarcinoma (NCT03374852; NCT01520805; and NCT03504410) [29,30]. However, it has not shown strong efficacy for these cancers as a single agent, possibly due to metabolic rewiring of malignant cells upon treatment [31,32]. Understanding how malignant cells escape pharmacological inhibition of the TCA cycle can facilitate the rational selection of combination therapy for CPI-613 and other TCA cycle inhibitors. 

We previously reported that a 50% reduction of DLST, the E2 subunit of KGDHC, can significantly delay MYC-driven leukemogenesis in zebrafish [33]. However, these lymphoblasts achieve renewed proliferation after a few months, leading to leukemia development in almost all of the heterozygous *Myc;dlst*+/− fish. Here, we study the effects of genetically depleting DLST to explore how T-ALL reprograms metabolism to escape TCA cycle disruption through genetic or pharmacological targeting. We demonstrate that T-ALL utilizes IDH-mediated reductive carboxylation to maintain the production of TCA cycle intermediates, which are needed to sustain leukemic cell survival and/or growth. We then provide experimental evidence that this rewiring of metabolic flux is linked to an epigenetically-mediated process of DNA demethylation driven by rising α-KG concentrations.

Our study provides novel insight into the biochemical process underlying T-ALL metabolic reprogramming and facilitates a rational selection of adjunctive metabolic therapy in combination with CPI-613 to treat MYC-driven T-ALL and perhaps other MYC-driven cancers as well.

## 2. Methods

### 2.1. Metabolomics Profiling and Analysis

Glutamine-free RPMI medium (11879020, Gibco, NY, USA) was supplemented with ^12^C or ^13^C isotope-labeled L-Glutamine (G8540 and 605166, Sigma, St. Louis, MI, USA) for experiments. Lentivirus transduction was performed as described in the following section. On day 3 post-transduction, cells were replenished with a 2 mM ^13^C or ^12^C glutamine medium. After 24 h, the cells were washed with Dulbecco’s Phosphate Buffered Saline (DPBS, SH30028LS, Corning, NY, USA) twice and extracted for polar metabolites.

Specifically, pre-chilled 80% methanol (MeOH, A412P-4, Fisher Scientific, NJ, USA) at −80 °C was added to the cells. After 20 min of incubation at −80 °C, the resulting mixture was collected into a 15-mL centrifuge tube and centrifuged for 5 min at 4000 rpm and 4 °C. Insoluble pellets were re-extracted with 1 mL of MeOH. The supernatants from two rounds of extraction were combined and then dried with a Savant DNA 120 speedVac (Thermo Scientific, Waltham, MA, USA). The unlabeled and ^13^C L-Glutamine labeled metabolites were analyzed by LC-MS at the Beth Israel Deaconess Medical Center in Boston, MA, USA.

### 2.2. Cell Culture and α-KG Treatment

MOLT3, PEER, and JURKAT cells were obtained from the DSMZ (https://www.dsmz.de/dsmz (accessed on 1 December 2003)). All T-ALL cells were cultured at 37 °C with 5% CO_2_ in RPMI-1640 medium (MT10040CV, Corning) supplemented with 10% fetal bovine serum (FBS, F0926, Sigma) except for PEER cells, which were supplemented with 20% FBS. T-ALL cells at the exponential growth phase were resuspended in fresh RPMI-1640 medium and seeded into a six-well plate with 2 million cells per well. Dimethyl α-KG (349631, Sigma) was added to the wells with the final concentrations of 0, 3.5, and 7 mM, respectively. Cells were treated for 12, 24, and 36 h or qPCR analysis and 24 and 48 h for western blotting analysis.

### 2.3. Protein Extraction and Western Blotting

Cells were lysed in RIPA buffer supplemented with 1x Halt proteinase inhibitor (87786, Thermo Scientific) and phosphatase inhibitor cocktail (BP479, Boston Bioproducts, Milford, MA, USA). The primary antibodies used include anti-DLST (H00001743, Abnova, Taipei, Taiwan), anti-IDH1 (LS-C497935, Life span bioscience, Seattle, WA, USA), anti-IDH2 (LS-C373619, Life span bioscience), anti-LDHA (#2012, Cell Signaling Technology, Danvers, MA, USA), and anti-ACTIN (sc-47778, Santa Cruz Biotechnology, Dallas, TX, USA). Secondary antibodies include anti-mouse (31430, Thermo Scientific) or anti-rabbit antibodies (65-6120, Thermo Scientific). Chemiluminescence Supersignal West Pico was from Thermo Scientific (Cat# 34080). Autoradiographs were imaged using a G: BOX Chemi XT4 (Syngene, Bengaluru, India).

### 2.4. Lentivirus Infection

A control and two *DLST* shRNA hairpins were cloned into a pLKO.1-puro vector: 5′-CTTCGAAATGTCCGTTCGGTT-3′ (*shLuciferase*), 5′-CCCTAGTGCTGGTATACTATA-3′ (*shDLST1*), and 5′-TGTCTCATAGCCTCGAATATC-3′ (*shDLST2*). Lentivirus production and transduction were conducted as previously described [11]. The media was changed 16 h after virus transduction, and puromycin (0.5 μg/mL, P7255, Sigma) was added 36 h post-transduction. On day 5 post-transduction, the cells were harvested, and western blotting was performed.

### 2.5. Immunofluorescence Staining and Imaging

Cells were cytospun, fixed, and permeabilized in cold acetone for 10 min and then air-dried. The cytospun cells were then washed with cold PBS for 5 min, denatured with 2N HCL for 15 min, neutralized with 100 mM Tris-HCl (pH 8.5) for 10 min RT, then washed with cold PBS, and incubated for 1 h with blocking buffer (10% horse serum and 5% bovine serum albumin in PBS containing 0.1% Triton X-100). Incubation was then performed with 5-hydroxymethylcytosine (5hmC; 40900, Active Motif, Carlsbad, CA, USA) primary antibodies (1:1000) in blocking buffer (5% horse serum and 1% bovine serum albumin in PBS containing 0.1% Triton X-100) overnight in a humidified chamber at 4 °C.

After three consecutive 10-min washes with PBS, cells were incubated with an Alexa Fluor 594–goat anti-rabbit secondary antibody (A32740, Thermo Scientific; 1:1000) in blocking buffer (5% horse serum and 1% bovine serum albumin in PBS containing 0.1% Triton X-100). The cells were washed again three times with PBS, followed by DAPI staining (D1306, Thermo Scientific; 1:10,000) for 10 min and mounting in a fluorescent mounting medium (VECTASHIELD Antifade Mounting Medium; H1900, Vector lab, Newark, CA, USA).

Images were captured using a Leica SP5 Confocal Microscope and LAS AF software. Sections were imaged at the same exposure settings and processed in identical conditions. Image quantification was performed using Fiji ImageJ software. The mean intensities were measured for about 10 random cell nuclei on each region of interest for each sample. The mean values of the mean intensities were plotted onto graphs. Experimental error is expressed as s.e.m.

### 2.6. Quantitative Real-Time PCR (qRT-PCR)

MOLT3, PEER, and JURKAT cells were cultured and transduced with *shLuciferase, shDLST1,* or *shDLST2* lentivirus as described above. On day 5 post-infection, the cells were harvested and subjected to total RNA extraction using Trizol reagent (15596026, Invitrogen). cDNA was synthesized with a Reverse Transcription Kit (205311, Qiagen, Hilden, Germany) for qPCR.

SYBR Green PCR master mix (QP004, Genecopoeia, Rockville, MD, USA) and a Step-One PCR instrument (Applied Biosystems, Waltham, MA, USA) were utilized for the qPCR reaction according to the manufacturer’s instructions. The qPCR primer sequences included: β-ACTIN (forward: 5′-GATTCCTATGTGGGCGACGA-3′, reverse 5′- AGGTCTCAAACATGATCTGGGT-3′), *IDH1* (forward: 5′- CTCTGTGGCCCAAGGGTATG-3′, reverse 5′-GGATTGGTGGACGTCTCCTG-3′), and *IDH2* (forward: 5′-ACAACACCGACGAGTCCATC-3′, reverse 5′-GCCCATCGTAGGCTTTCAGT-3′. All reactions were performed in triplicate.

### 2.7. Zebrafish Xenograft Assays

Zebrafish (*Danio rerio*) husbandry was performed as described in the aquatic facility at Boston University School of Medicine (BUSM), following the protocols approved by the Institutional Animal Use and Care Committee at BUSM.

PEER cells were stained with cell tracker dye (C7001, Thermo Fisher, Waltham, MA, USA) according to the manufacturer’s instructions one day before the xenograft. Zebrafish embryos were gradually reared to 37 °C and injected with ~200 PEER cells into the perivitelline space of each embryo at two days post-fertilization (dpf). For the drug treatment experiment, fish embryos were treated with vehicle, 2 μM CPI-613 (HY-15453, MCE, NJ, USA), 4 μM Bobcat339 (HY-111558A, MCE), or a combination of 2 μM CPI-613 and 4 μM Bobcat339. At 48 h after treatment, fish were observed under the microscope and their images were captured.

### 2.8. Statistical Analysis

Student’s *t*-test was used to analyze differences in ^13^C labeled TCA cycle metabolites between the control and *DLST* knockdown groups. One-way analysis of variance (ANOVA) was utilized to assess differences in *IDH1* and *IDH2* mRNA expression, 5hmC intensity, and zebrafish xenograft tumor burden. *p* values equal to or less than 0.05 were considered to be statistically significant.

### 2.9. Timeline Proposed to Complete Future Experiments

We plan to complete the required experiments within 6 months after the completion of the Stage 1 review.

## 3. Results

### 3.1. DLST Knockdown in Human T-ALL Cells Enhances Reductive Carboxylation While Decreasing Glutamine-Derived TCA Cycle Flux

We previously reported that depleting DLST, a TCA cycle transferase, failed to prevent leukemogenesis despite a delayed tumor onset, raising the question of how T-ALL cells overcome TCA cycle dysfunction to meet their metabolic needs [11]. To address this question, we knocked down *DLST* by shRNA and performed targeted metabolomic profiling in T-ALL cells that were labeled with ^13^C glutamine as previously described (Figure 1A) [34]. Over 90% of the labeled glutamine was incorporated in T-ALL cells with *shDLST* or *shLuciferase* transduction, without significant differences detected (Figure 1B).

As expected, LC-MS analysis demonstrated that DLST depletion significantly decreased the production of cycle intermediates derived from ^13^C glutamine, as demonstrated by reduced percentages of M + 2 fumarate, malate, aspartate, and citrate, as well as M + 4 citrate (Figure 1E–H). In contrast, we observed increased reductive carboxylation, shown by significantly increased percentages of M + 3 fumarate, malate, and aspartate, as well as M + 5 citrate (Figure 1E–H). Taken together, *DLST* inactivation in T-ALL cells decreases TCA cycle flux while increasing reductive carboxylation.

### 3.2. DLST Knockdown Leads to Upregulation of IDH1 and IDH2 in Human T-ALL Cells

Previous research has established the role of IDH1 and/or IDH2 in mediating reductive carboxylation in acute myeloid leukemia (AML) cells [18,35]. To understand the biochemical mechanism facilitating the shift from KGDHC-mediated oxidative decarboxylation to reductive carboxylation, we detected IDH expression in a panel of human T-ALL cell lines in the presence or absence of *DLST* inactivation. qRT-PCR analysis showed that DLST inactivation significantly increased transcript levels of *IDH1* and *IDH2* in human PEER and JURKAT T-ALL cell lines (Figure 2A,B).

Interestingly, we only detected increased transcript levels of *IDH1* but not *IDH2* in human MOLT3 T-ALL cells (Figure 2A,B). Next, we asked if the increased transcription of *IDH1* and *IDH2* would translate to increased protein levels. Western blotting analysis revealed a general trend of increased protein levels of IDH1 and IDH2 among all three T-ALL cell lines despite some fluctuations (Figure 2C). The original western blot images for Figure 2C are included in Appendix A. Together, these results demonstrate that TCA cycle disruption by DLST inactivation leads to upregulation of both transcript and protein levels of IDH1 and IDH2, enzymes critical for reductive carboxylation.

### 3.3. Supplementation of Exogenous α-KG Induces Upregulation of IDH1 and IDH2 in Human T-ALL Cells

Having established that TCA inhibition causes T-ALL to upregulate IDH1 and IDH2 protein expression via increased transcription, we sought to determine whether a buildup of TCA cycle metabolic intermediates was acting as a trigger for this. We previously reported that DLST depletion in human T-ALL cells led to a significant increase of α-KG, which is the substrate of KGDHC and through which glutamine enters the TCA cycle [11]. To investigate the mechanism by which TCA cycle disruption induces upregulation of IDH1 and IDH2, we treated three human T-ALL cell lines with exogenous α-KG to mimic the effect of KGDHC dysfunction due to genetic depletion of DLST.

Specifically, MOLT3, PEER, and JURKAT cells were incubated with 0, 3.5, and 7 mM of α-KG for 12, 24, and 36 h. We then measured the transcript levels of *IDH1* and *IDH2* by qRT-PCR for these cells. We detected a concentration-dependent increase in *IDH1* and *IDH2* mRNA levels upon α-KG treatment (Figure 3A,B). Similarly, western blotting analysis of the three T-ALL cell lines detected increased the protein levels of IDH1 and IDH2 upon α-KG treatment for 24 and 48 h (Figure 3C). The original western blot images for Figure 3C are included in Appendix A. Taken together, exogenous α-KG treatment mimics genetic disruption of the TCA cycle by DLST inactivation, leading to increased transcript and protein levels of IDH1 and IDH2 in human T-ALL cells.

### 3.4. Both α-KG Treatment and DLST Depletion Lead to Increased DNA Demethylation in Human T-ALL Cells

Besides serving as a cycle intermediate, α-KG is the obligatory co-factor for multiple α-KG-mediated dioxygenases, such as the TET family of DNA demethylases [36,37]. Changes in the methylation state of DNA indicate epigenetic alterations that can regulate gene expression in leukemic and other cancer cells, allowing them to hijack or reprogram cellular processes to favor growth and proliferation [38,39]. Hence, we sought to determine whether the increase in *IDH1/2* transcription under DLST knockdown or α-KG supplementation was due to an epigenetic change in the methylation state of T-ALL DNA.

We performed immunofluorescent staining of 5hmC (a marker of DNA demethylation) after treating human MOLT3 and PEER cells with 0, 3.5, and 7 mM of α-KG for 24 and 48 h. DAPI counterstain of nuclei revealed that α-KG treatment led to a significant increase of 5hmC staining in both cell lines (Figure 4A,B). Next, we determined if DLST depletion can also increase 5hmC levels. Similar to α-KG treatment, *DLST* knockdown led to a significant increase of DNA demethylation in both MOLT3 and PEER cells, as demonstrated by increased 5hmC staining (Figure 5A,B). Together, our results show that both α-KG treatment and TCA cycle disruption via DLST depletion lead to increased DNA demethylation in human T-ALL cells, linking upregulated *IDH* expression with α-KG-mediated demethylation.

### 3.5. Combined Treatment with CPI-613 and Bobcat339 Significantly Decreases Tumor Burden in Zebrafish Xenografts of Human T-ALL

Having established a link between α-KG-mediated demethylation and upregulation of IDH1/2, which promotes reductive carboxylation, we sought to determine the importance of α-KG-mediated demethylation in T-ALL maintenance. Pharmacological inhibition of the DNA demethylase, TET1 and TET2, by BobCat339 somehow reduced the T-ALL burden in zebrafish xenografts (Figure 6A,B). Inhibiting the TCA cycle by CPI-613, an inhibitor of pyruvate dehydrogenase and mitochondrial KGDHC, showed similar trends (Figure 6A,B).

However, none of the single-agent treatments were able to significantly reduce the tumor burden. Importantly, the combination treatment of CPI-613 and BobCat339 showed an additive effect, leading to a significantly reduced tumor burden in zebrafish xenografts, as demonstrated by the quantification of fluorescence intensity of T-ALL cells (Figure 6A,B). Our findings demonstrate that epigenetic inhibition in combination with single-agent targeting of the TCA cycle can overcome drug resistance caused by tumor metabolic plasticity.

## 4. Discussion

Metabolic rewiring driven by oncogenes presents unique opportunities for therapeutic intervention, although challenges remain in overcoming treatment-induced metabolic plasticity [1,3,40]. The proto-oncogene *MYC* is a central regulator of cellular metabolism and is deregulated in over half of human cancers, including T-ALL, which relies on the TCA cycle and glutamine anaplerosis [4,5,6,7,11]. Given that MYC-driven glutamine anaplerosis produces cycle intermediates for macromolecule synthesis, cycle inhibition is expected to impair the production of glutamine-derived carbon skeletons in T-ALL [14,41,42].

However, we found that reductive carboxylation helps maintain glutamine uptake and production of cycle intermediates despite KGDHC disruption. This correlates with our prior work showing that T-ALL cells with depletion of the cycle transferase DLST resume tumorigenesis after a delayed onset [11]. Therefore, isolated cycle inhibition may not be an effective therapeutic approach due to metabolic rescue via reductive carboxylation. Identifying mechanisms promoting reductive carboxylation will enable the rational selection of combination therapy to achieve maximal treatment efficacy.

Our results uncovered one novel mechanism by which cancer cells increase reductive carboxylation in response to mitochondrial dysfunction. We show that DLST depletion in T-ALL cells, which disrupts KGDHC function and leads to α-KG accumulation [11], increased mRNA and protein levels of IDH1 and IDH2, the enzymes responsible for reductive carboxylation. Importantly, exogenous α-KG treatment had the same effects as DLST depletion, suggesting that α-KG is the key mediator in regulating IDH expression.

Our observation of increased 5hmC levels under both α-KG treatment and KGDHC disruption in T-ALL suggests that increased α-KG upon KGDHC dysfunction subsequently promotes TET-induced DNA demethylation. Hence, the transcription of *IDH* mRNA is most likely induced by α-KG-mediated TET demethylation of DNA.

Different from the present work, prior studies did not examine cellular transcriptional changes as the mechanism to increase reductive carboxylation upon mitochondrial dysfunction. Rather, they explored three other mechanisms: the effects of increased NADPH levels—on which reductive carboxylation is dependent; the role of hypoxia-inducible factors in increasing glutamine metabolism through reductive carboxylation; or reductive carboxylation induced by anchorage loss [17,18,33,43,44].

Interestingly, our study demonstrates that epigenetic changes are linked to the increased reductive carboxylation in T-ALL in response to mitochondrial dysfunction. Furthermore, we showed that both 5hmC fluorescence intensity and IDH1/2 expression underwent progressive increases with rising α-KG concentrations, indicating a direct relationship between accumulation of α-KG, demethylation of DNA, and IDH1/2 expression. Taken together, our data indicate that disruption of the TCA cycle leads to α-KG accumulation, which activates TET demethylases to upregulate IDH mRNA and protein, thereby, leading to increased reductive carboxylation to compensate for compromised TCA cycle function (Figure 7A).

While OXPHOS dysfunction has been shown to induce IDH1/2-mediated reductive carboxylation in certain cancer cells, the TCA cycle function is intact in this context [17]. The intact TCA cycle allows the production of NADH and NADPH to support reductive carboxylation, and KGDHC is required for maximum activity of reductive carboxylation [17]. However, our targeted metabolomics demonstrated increased reductive flux despite the significant disruption of the TCA cycle via depletion of the KGDHC E2 subunit DLST.

NADH levels appear to be further reduced beyond the effect of KGDHC inhibition, as demonstrated by significant decreases in percentages of M + 4 citrate and M + 1 α-KG, both of which are downstream of the two other NADH-producing cycle reactions. Our studies show that there is an increased reductive carboxylation despite the loss of KGDHC and TCA cycle-derived reducing equivalents, a provocative finding given the previously established dependence of reductive carboxylation on oxidative TCA cycle flux [17].

Dysfunction of TET enzymes has been associated with tumorigenesis in both solid and hematological malignancies [45,46]. In AML, TET inhibition frequently occurs due to aberrant production of the oncometabolite 2-HG, which leads to hypermethylated promoter regions, possibly contributing to the silencing of tumor suppressors [47,48]. As a result, a variety of compounds that seek to increase TET activity, including IDH inhibitors and TET activators, have been developed [47]. However, TET enzymes are believed to play a multifaceted and context-specific role during tumorigenesis, as the methylation state of malignant genomes can be both hyper- and hypomethylated [49].

Our results support that T-ALL utilizes TET-mediated demethylation to promote survival, likely through enhancing IDH1/2 expression and reductive carboxylation in response to inhibition of central carbon metabolism. Supporting the role of DNA demethylation in T-ALL survival, combination treatment with a TCA cycle inhibitor (CPI-613) and a TET1/2 inhibitor (Bobcat339) resulted in an additive effect in reducing the T-ALL burden. These data suggest that pharmacologic inhibition of the TCA cycle also forces T-ALL to utilize a TET-dependent metabolic reprogramming, e.g., reductive carboxylation, which can be therapeutically exploited. While the metabolic plasticity of T-ALL may allow them to escape isolated TCA cycle inhibition, the combined inhibition of TET prevents activation of alternative metabolic pathways, thus, leading to increased therapeutic efficacy (Figure 7B).

We have now demonstrated, for the first time, that direct TCA inhibition can drive reductive carboxylation in cancer cells. Hence, direct TCA cycle inhibitors, such as CPI-613, should be grouped with Electron Transport Chain inhibitors, such as metformin, rotenone, and antimycin, which can force cancer cells to rely on reductive carboxylation. However, using these inhibitors as a single-agent treatment poses clinical challenges given the capacity of cancer cells to rely on reductive carboxylation for survival, making the identification of how the reductive pathway is activated and maintained a priority.

We demonstrate that reductive carboxylation in T-ALL is likely activated via TET-dependent DNA demethylation, pinpointing a novel combined therapeutic approach. This finding may have broader implications for other MYC-driven cancers with shared metabolic derangements, especially given our recent findings that neuroblastoma and subsets of TNBC share a similar dependency to T-ALL on DLST [27,28]. However, given the context-specific nature of MYC-driven metabolic alterations and TET activity, the contribution of α-KG-mediated DNA demethylation to reductive carboxylation should first be established in other cancers before pharmacologic studies are undertaken.

## 5. Conclusions

Combining targeted metabolomics profiling with molecular and biochemical studies, we showed that human T-ALL rewires its metabolism from oxidative decarboxylation to reductive carboxylation through the enhanced transcription of IDH1/2. Supporting the transcriptional increase of IDH1/2, we detected an increase in DNA demethylation driven by rising α-KG concentrations. Consequently, dual pharmacologic inhibition of the TCA cycle and TET demethylases demonstrated additive efficacy in significantly reducing the T-ALL burden in zebrafish xenografts. Hence, reductive carboxylation induced by epigenetic DNA modifications sustained T-ALL survival upon TCA cycle targeting, and combination treatment with CPI-613 and TET inhibitors could improve treatment responses.

## Figures and Tables

**Figure 1 cancers-14-02983-f001:**
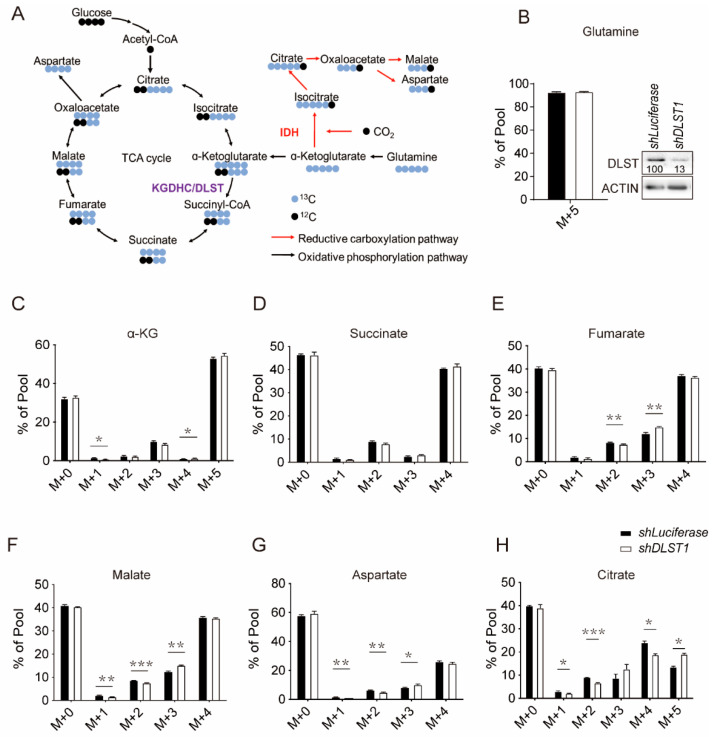
*DLST* inactivation increased reductive carboxylation while reducing TCA cycle intermediates in human T-ALL cells. (**A**) Scheme of ^13^C labeled glutamine contribution to TCA cycle metabolites. (**B**) Percentage of ^13^C-labelled glutamine uptake in human MOLT3 cells in the presence or absence of *DLST* knockdown. The insert in panel (**B**) shows efficient depletion of DLST by *shDLST1* hairpin. (**C**–**H**) Percentages of TCA cycle metabolites derived from ^13^C labeled glutamine in the presence or absence of *DLST* knockdown. M stands for mass. The labels for M + 0, M + 1, M + 2, M + 3, M + 4, and M + 5 indicate that no or one to five carbons in the metabolite are derived from ^13^C labeled glutamine. The data are presented as the mean ± s.e.m. and analyzed with an unpaired two-tailed *t*-test for statistical significance. * *p* < 0.05, ** *p* < 0.01, *** *p* < 0.001.

**Figure 2 cancers-14-02983-f002:**
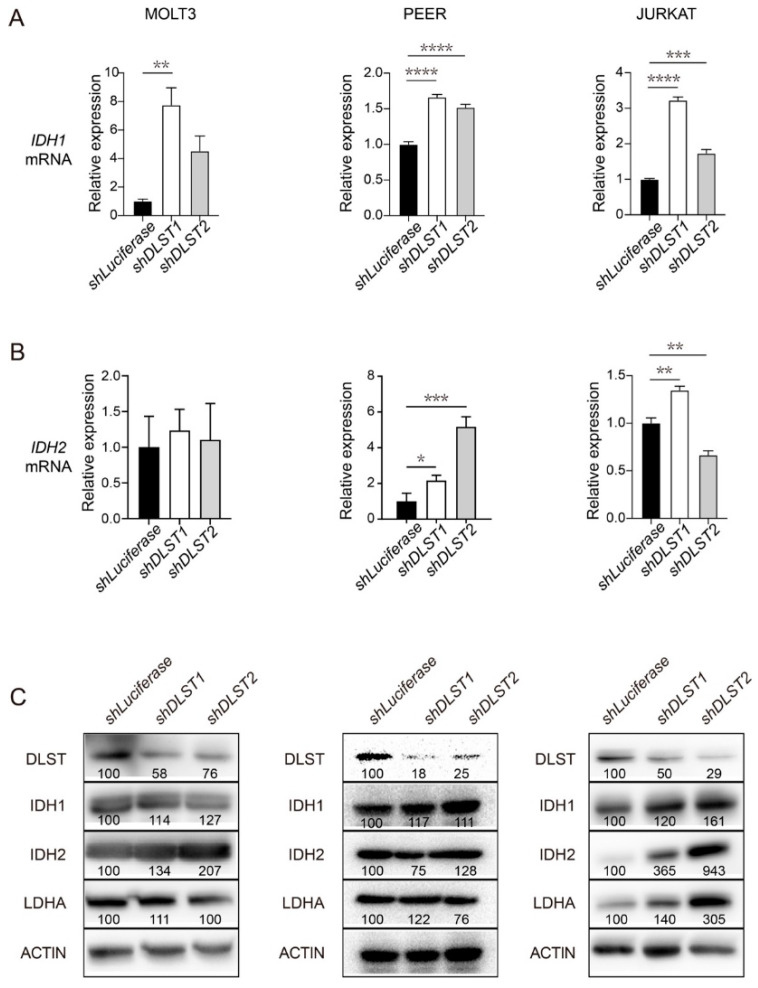
*DLST* inactivation increases transcript and protein levels of IDH1 and IDH2 in human T-ALL cells. (**A**,**B**) qRT-PCR analysis of *IDH1* (**A**) and *IDH2* (**B**) transcript levels at 4 days transduction of *shLuciferase* and *shDLST* in human MOLT3, PEER, and JURKAT T-ALL cell lines. Relative fold change was analyzed using the *shLuciferase* group as a control. The data are presented as the mean ± s.e.m, and statistical differences were calculated using one-way ANOVA. (**C**) Western blotting of DLST, IDH1, IDH2, and LDHA in the same three T-ALL cell lines transduced with *shLuciferase* or *shDLST* hairpins. Relative protein to ACTIN ratios are shown at the bottom of each western blotting panel. * *p* < 0.05, ** *p* < 0.01, *** *p* < 0.001, **** *p* < 0.0001.

**Figure 3 cancers-14-02983-f003:**
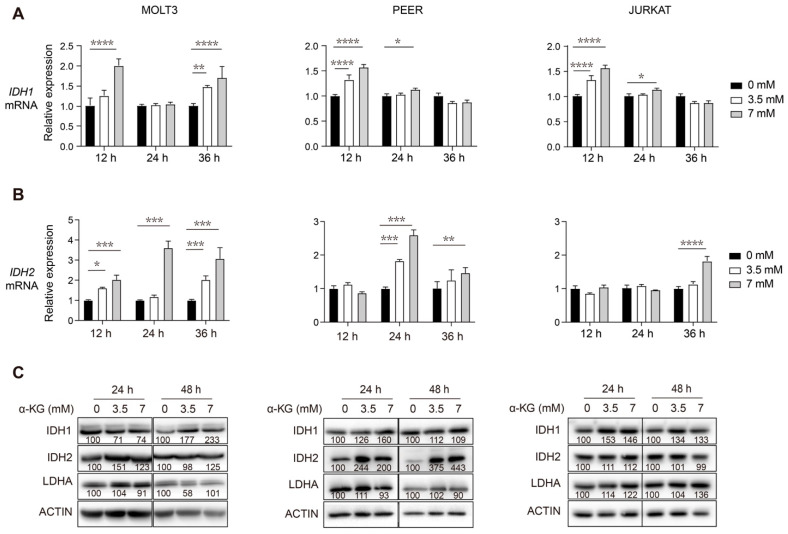
α-KG treatment of human T-ALL cells leads to increased transcript and protein levels of IDH1 and IDH2. (**A**,**B**) qRT-PCR analysis of *IDH1* (**A**) and *IDH2* (**B**) transcript levels at 12, 24, and 36 h post-treatment with 0, 3.5, or 7 mM of α-KG in human MOLT3, PEER, and JURKAT T-ALL cell lines. Relative fold change was analyzed using the 0 mM group as a control. The data are presented as the mean ± s.e.m, and statistical differences are calculated and normalized to the control, and determined by two-way ANOVA. (**C**) Western blotting of DLST, IDH1, IDH2, and LDHA in the same three T-ALL cell lines after 24 and 48 h of treatment with either 0, 3.5, or 7 mM of α-KG. Relative protein to ACTIN ratios are shown at the bottom of each western blotting panel. * *p* < 0.05, ** *p* < 0.01, *** *p* < 0.001, **** *p* < 0.0001.

**Figure 4 cancers-14-02983-f004:**
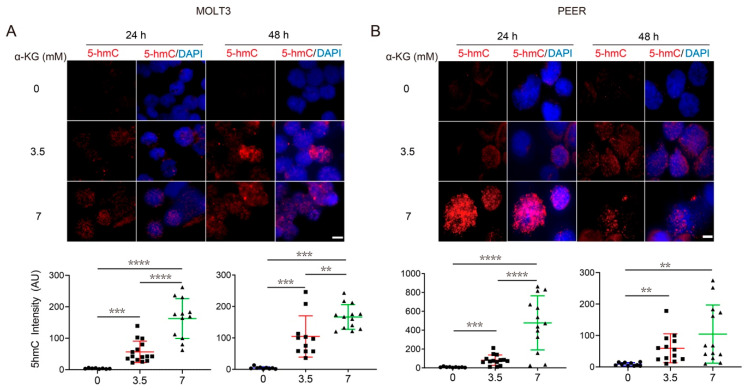
α-KG treatment in T-ALL cells increases DNA demethylation in a concentration-dependent manner. (**A**,**B**) 5hmC staining in MOLT3 (**A**) and PEER (**B**) cells after 24 and 48 h of treatment with 0, 3.5, or 7 mM of α-KG. DAPI (blue) counterstaining indicates the intranuclear location. The quantification of relative 5hmC fluorescence (red) is presented as the mean ± s.e.m., and an unpaired two-tailed *t*-test is used for statistical analysis. Scale bar = 10 μm. ** *p* < 0.01, *** *p* < 0.001, **** *p* < 0.0001.

**Figure 5 cancers-14-02983-f005:**
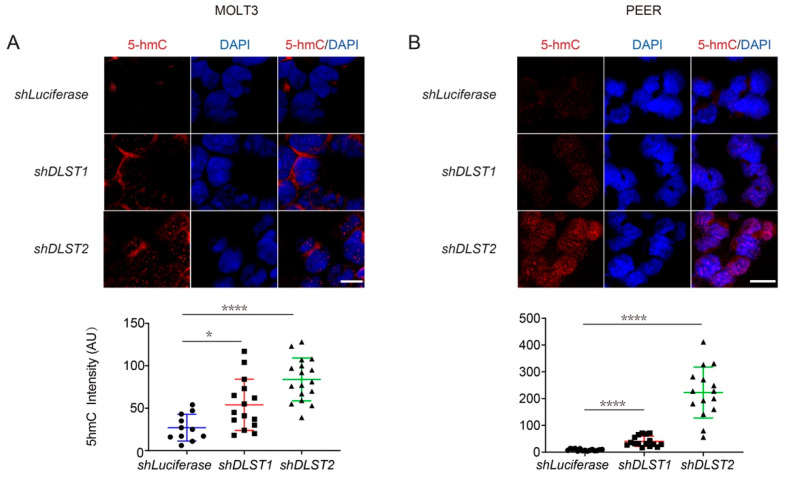
*DLST* inactivation significantly increases DNA demethylation in human T-ALL cells. (**A**,**B**) 5hmC staining in MOLT3 (**A**) and PEER (**B**) cells at 5 days after *DLST* knockdown, with *Luciferase* knockdown as the control and DAPI (blue) counterstain to demonstrate the intranuclear location of 5hmC staining (red). The quantification of relative 5hmC fluorescence intensity is presented as the mean ± s.e.m., and statistical differences are calculated using an unpaired two-tailed *t*-test. Scale bar = 10 μm. * *p* < 0.05, **** *p* < 0.0001.

**Figure 6 cancers-14-02983-f006:**
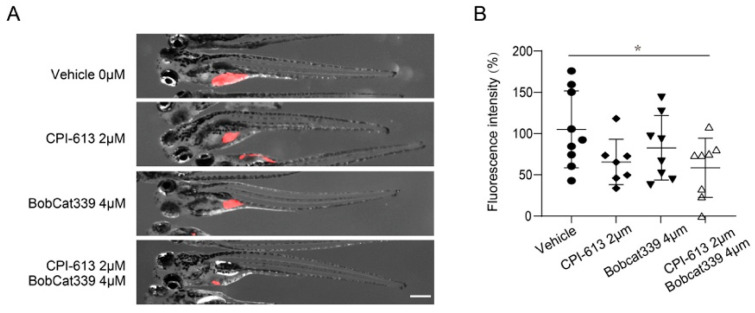
Combined inhibition of the TCA cycle and DNA methylation through CPI-613 and Bobcat 339 significantly decreases the tumor burden in zebrafish xenografts of human T-ALL. (**A**) Treatment of zebrafish xenografts with the TCA cycle inhibitor CPI-613 (2 μM) or TET1/2 inhibitor Bobcat339 (4 μM) resulted in a slightly decreased tumor burden, while combination treatment led to a significantly decreased tumor burden. (**B**) Quantification of the tumor fluorescence intensity in zebrafish xenografts is shown in (**A**). The data are presented as the mean ± s.e.m., and statistical differences are calculated using an unpaired two-tailed *t*-test with the vehicle as the control. Scale bar in (**A**) = 500 μm. * *p* < 0.05.

**Figure 7 cancers-14-02983-f007:**
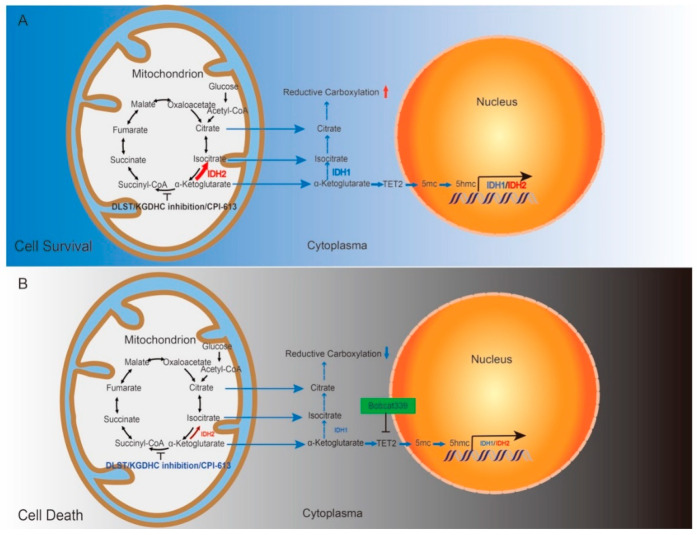
Schematic drawing of our working model. (**A**) Disrupting the TCA cycle alone leads to enhanced reductive carboxylation and T-ALL cell survival through TET-mediated demethylation and upregulation of IDH1 and IDH2. (**B**) Combined inhibition of the TCA cycle and TET-mediated demethylation kills T-ALL cells by decreasing the production of IDH1 and IDH2 and subsequently dampening reductive carboxylation.

## Data Availability

The original western blot images are in Appendix A. The rest of the original data are available upon request.

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
