# Peer review of "α-Ketoglutarate-Mediated DNA Demethylation Sustains T-Acute Lymphoblastic Leukemia upon TCA Cycle Targeting"

_cancers, 2022, doi:10.3390/cancers14122983_

Round 1

Reviewer 1 Report

In this manuscript, Authors analyzed α-ketoglutarate-mediated DNA demethylation sustains T-acute lymphoblastic leukemia upon TCA cycle targeting.

This is an interesting review. 

I suggest the Authors consider moving Figure 7 to the Results section.

Author Response

Review comments: In this manuscript, Authors analyzed α-ketoglutarate-mediated DNA demethylation sustains T-acute lymphoblastic leukemia upon TCA cycle targeting.

This is an interesting review. I suggest the Authors consider moving Figure 7 to the Results section.

Response: We appreciate the Reviewer for the suggestion. We have moved Figure 7 to the end of the Result section.

Reviewer 2 Report

Wang et al, in their manuscript "α-ketoglutarate-mediated DNA demethylation sustains T-acute lymphoblastic leukemia upon TCA cycle targeting", 

They show that human T- acute lymphoblastic leukemia (T-ALL) cells shift their metabolism from oxidative decarboxylation to reductive carboxylation when the tricarboxylic acid (TCA) cycle is disrupted. 
By metabolomics they demonstrated increased reductive carboxylation after genetic depletion of the E2 subunit of KGDHC, acting as pharmacological inhibition of the complex. KGDHC dysfuntion increases demethylation of nuclear DNA by α-KG-dependent dioxygenases and augments IDH1 and 2. Pharmacologic inhibition of both the TCA cycle and TET demethylases demonstrates synergic efficacy as antitumoral activity in zebrafish xenografts. The findings shed light in mechanisms of T-ALL resistance to TCA cycle targeting  drugs and suggest new therapeutic strategies. 
The methods appear sound and results are well described. Pictures are clear and appropriately explained
The findings may provide a basis for future studies and development of new therapeutic strategies in T ALL

Author Response

Reviewer comments: Wang et al, in their manuscript "α-ketoglutarate-mediated DNA demethylation sustains T-acute lymphoblastic leukemia upon TCA cycle targeting", They show that human T- acute lymphoblastic leukemia (T-ALL) cells shift their metabolism from oxidative decarboxylation to reductive carboxylation when the tricarboxylic acid (TCA) cycle is disrupted.

By metabolomics they demonstrated increased reductive carboxylation after genetic depletion of the E2 subunit of KGDHC, acting as pharmacological inhibition of the complex. KGDHC dysfuntion increases demethylation of nuclear DNA by α-KG-dependent dioxygenases and augments IDH1 and 2. Pharmacologic inhibition of both the TCA cycle and TET demethylases demonstrates synergic efficacy as antitumoral activity in zebrafish xenografts. The findings shed light in mechanisms of T-ALL resistance to TCA cycle targeting  drugs and suggest new therapeutic strategies.

The methods appear sound and results are well described. Pictures are clear and appropriately explained. The findings may provide a basis for future studies and development of new therapeutic strategies in T ALL

Response: We thank the Reviewer for the favorable comments.

Reviewer 3 Report

In the current research article, Wang, Shen and colleagues explore the metabolic rewiring that occurs in T-ALL when the TCA cycle is disrupted. This is particularly interesting since single-agent therapeutic approaches targeting the TCA have been quite unsuccessful and therefore call for combination therapies.
Within this study the authors show that the knockdown of KGDHC/DLST, a key enzyme in the TCA cycle, shifts the balance from the oxidative phosphorylation pathway to the reductive carboxylation pathway. This is accompanied by the increased expression of IDH1/IDH2, which fuel the production of TCA intermediates and promote cell survival. A similar result is obtained when exogenous alpha-KG is administered to the cells in culture. A combined blockade of the two pathways has an effect in vivo in a T-ALL zebrafish xenograft model.

Overall, the article is well written and explained. In particular, the schemes in fig 1a and 7 are helpful to the reader to follow the discussion on metabolic pathways. In spite of a good and intriguing message, I feel that a few crucial aspects have been overlooked, therefore I would suggest the authors to expand on these. 

One of the main messages of the study is that DLST depletion (and alpha-KG increase) cause an increase in DNA methylation, which calls upon an increased activity of TET demethylases. The authors show that DLST depletion/a-KG supplementation cause a global increase of DNA demethylation (5hmC) in 3 different T-ALL cell lines, but don’t touch on a very important side of the story. Since TET enzymes have a context-specific role, acting a variety of genes (including tumor suppressors), the authors should show where (i.e. in which specific regions) DNA demethylation occurs upon DLST depletion, therefore assessing whether cell cycle genes or resistance genes are affected. This can be easily done by bisulphite sequencing or by TET2 ChIP-sequencing.
Additionally, since demethylation increases accessibility of specific genomic regions, a chromatin accessibility profiling (ATAC-seq) upon DLST depletion would give a better picture of the genome-wide results.
Of course I understand that this additional and more expensive effort could move the paper to a higher-tier journal, so I leave to the authors to decide which strategy to chose for this study.

Minor points

The first figure of the paper shows the metabolic consequences of DLST knockdown by shRNA. The authors should include a figure (even as supplementary info) to show the percentage of silencing (RNA and protein levels) upon treatment with shRNA for DLST or controls. 

Also, they should include an explanation of the annotations M+0, M+1, M+2 etc in figure 1 legend. 

In the discussion, ETC inhibitors are mentioned but the acronym is not spelled out.

Author Response

Point 1: One of the main messages of the study is that DLST depletion (and alpha-KG increase) cause an increase in DNA methylation, which calls upon an increased activity of TET demethylases. The authors show that DLST depletion/a-KG supplementation cause a global increase of DNA demethylation (5hmC) in 3 different T-ALL cell lines, but don’t touch on a very important side of the story. Since TET enzymes have a context-specific role, acting a variety of genes (including tumor suppressors), the authors should show where (i.e. in which specific regions) DNA demethylation occurs upon DLST depletion, therefore assessing whether cell cycle genes or resistance genes are affected. This can be easily done by bisulphite sequencing or by TET2 ChIP-sequencing.

Additionally, since demethylation increases accessibility of specific genomic regions, a chromatin accessibility profiling (ATAC-seq) upon DLST depletion would give a better picture of the genome-wide results.

Of course I understand that this additional and more expensive effort could move the paper to a higher-tier journal, so I leave to the authors to decide which strategy to chose for this study.

Response 1: We agree with the Reviewer that the identification of specific gene regions affected by TET-mediated DNA demethylation could provide additional information. However such experiments would require a substantial investment of time and resources and would significantly delay the publication of this manuscript. Therefore, we have elected to delay these studies and believe that our current data are publication-worthy due to making a novel contribution to the existing literature. Additional studies with ChIP-sequencing or ATAC-seq would be a fruitful area for future investigation.

Point 2: The first figure of the paper shows the metabolic consequences of DLST knockdown by shRNA. The authors should include a figure (even as supplementary info) to show the percentage of silencing (RNA and protein levels) upon treatment with shRNA for DLST or controls.

Response 2: We apologize for the oversight. We have now included the western blotting data in the revised Figure 1 to demonstrate the efficient knockdown of DLST in human MOLT3 T-ALL cells. We agree with the Reviewer that it would be ideal to show RNA levels upon shDLST knockdown. However, we did not preserve cells for this experiment so cannot perform RNA extraction.

Point 3: Also, they should include an explanation of the annotations M+0, M+1, M+2 etc in figure 1 legend.

Response 3: We have included an explanation of the annotations in Figure 1 legend.

“M stands for mass. The labels for M+0, M+1, M+2, M+3, M+4, and M+5 indicate that no or 1 to 5 carbons in the metabolite are derived from 13C labeled glutamine.”

Point 4: In the discussion, ETC inhibitors are mentioned but the acronym is not spelled out.

Response 4: We have spelled out the acronym for ETC inhibitors.

“Electron Transport Chain inhibitors”

Round 2

Reviewer 3 Report

The authors addressed most of my comments in a satisfactory manner for the standard of the current publication. Overall, is an interesting study and I am curious to see its future developments.